# Center of Pressure Measurement Accuracy via Insoles with a Reduced Pressure Sensor Number during Gaits

**DOI:** 10.3390/s24154918

**Published:** 2024-07-29

**Authors:** Philip X. Fuchs, Wei-Han Chen, Tzyy-Yuang Shiang

**Affiliations:** 1Department of Athletic Performance, National Taiwan Normal University, No. 88, Section 4, Tingzhou Road, Wenshan District, Taipei City 116, Taiwan; gn01800083@gmail.com (W.-H.C.); tyshiang@ntnu.edu.tw (T.-Y.S.); 2Department of Physical Education and Sport Sciences, National Taiwan Normal University, No. 162, Section 1, Heping East Road, Da’an District, Taipei City 106, Taiwan; 3Department of Physical Education and Kinesiology, National Dong Hwa University, No. 1, Section 2, Da Hsueh Road, Shoufeng District, Hualian City 974301, Taiwan

**Keywords:** analysis, assessment, biomechanics, kinetics, technology

## Abstract

The objective was to compare simplified pressure insoles integrating different sensor numbers and to identify a promising range of sensor numbers for accurate center of pressure (CoP) measurement. Twelve participants wore a 99-sensor Pedar-X insole (100 Hz) during walking, jogging, and running. Eight simplified layouts were simulated, integrating 3–17 sensors. Concordance correlation coefficients (CCC) and root mean square errors (RMSE) between the original and simplified layouts were calculated for time-series mediolateral (ML) and anteroposterior (AP) CoP. Differences between layouts and between gait types were assessed via ANOVA and Friedman test. Concordance between the original and simplified layouts varied across layouts and gaits (CCC: 0.43–0.98; $\chi_{\left(7\right)}^{2}$ ≥ 34.94, *p* < 0.001). RMSE_ML_ and RMSE_AP_ [mm], respectively, were smaller in jogging (5 ± 2, 15 ± 9) than in walking (8 ± 2, 22 ± 4) and running (7 ± 4, 20 ± 7) ($\eta_{p}^{2}$: 0.70–0.83, *p* < 0.05). Only layouts with 11+ sensors achieved CCC ≥ 0.80 in all tests across gaits. The 13-sensor layout achieved CCC ≥ 0.95 with 95% confidence, representing the most promising compromise between sensor number and CoP accuracy. Future research may refine sensor placement, suggesting the use of 11–13 sensors. For coaches, therapists, and applied sports scientists, caution is recommended when using insoles with nine or fewer sensors. Consulting task-specific validation results for the intended products is advisable.

## 1. Introduction

Centre of pressure (CoP) is the most analyzed characteristic in studies of postural control [1]. It can be used to assess the risk of falling [2], detect pathologies [3], differentiate between foot arch types [4], and monitor training effects on balance performance [5]. Since maintaining equilibrium is also a requirement in dynamic activities, CoP plays an important role in gait analyses [6,7]. In-shoe systems (e.g., insoles equipped with pressure sensors) are frequently used [8] and recommended [9] tools for measuring CoP.

Mobile and wireless insoles are available [8] and allow for participants’ free movement during spacious tasks with almost no site restrictions. One limitation is the large number of sensors that is incorporated in many commercial products. For example, the well-established and recommended Pedar-X insole [10,11,12] comprises 99 sensors. This creates high product costs and amounts of data that limit their usage in clinical and therapeutical settings [13]. Since in-shoe systems are potentially valuable tools for diagnosis and monitoring in practical fields, low-cost alternatives with fewer sensors are needed [14,15]. Reducing the sensor number may help to reduce product costs, increase operating time due to reduced data amount, and improve the ease of use [13]. Besides other limiting factors (e.g., placement of device components), the data amount and therefore operating time (25 min to 6 h) is a limitation of great relevance for sports applications and can be influenced by sensor number [13]. However, lower sensor number may reduce CoP accuracy as documented previously [16]. Therefore, the relationship between sensor number and measurement accuracy should be investigated to facilitate the adoption of in-shoe systems in practical fields.

In any attempt of sensor reduction, the sensor placement plays an important role and should be in accordance with the foot anatomy and pressure distribution [8]. Previous research divided the foot into 10 different regions with 60.5% of the weight observed in the heel, 28.1% in the forefoot, 7.8% in the midfoot, and 3.6% in the toes [17]. Other 9-region profiles differ only marginally, merging two midfoot [18] or tarsal [19] regions because of the limited pressure measured in these regions [17]. Therefore, the 9 regions of the foot can be defined as medial heel, lateral heel, midfoot, medial forefoot, central forefoot, lateral forefoot, hallux, second to third tarsal, and fourth to fifth tarsal. These regions should be covered by sensor layouts of in-shoe pressure systems and were considered in most research that introduced simplified insoles with reduced sensor numbers.

Multiple studies evaluated prototypes and commercial products with reduced sensor numbers. One such product is the OpenGO insole comprising 13 sensors across the 9 regions of the foot. The OpenGO insole was recommended as a valid and reliable tool for various movements but not for unilateral balance tests due to large errors observed during single-leg balance and Y-Excursion tests [16]. Those tests have been the only tasks where the previous authors [16] calculated CoP. Other tasks such as gaits and jumps achieved better accuracy, based on vertical ground reaction forces (vGRF) [16]. This suggests that the sensor number may affect vGRF differently than CoP. This difference may be expected because a lower sensor number reduces measurement resolution, which is required to determine the accurate location of CoP. Moreover, it suggests that accuracy cannot be generalized for different movements, which was confirmed for vGRF during different types of locomotion [20]. During locomotion, the OpenGO showed larger relative bias than the 99-sensor Pedar-X [16], which was recognized as a highly accurate pressure insole [10,11,12].

A 12-sensor and a 9-sensor layout were tested via Spearman’s correlation (>0.95) and root mean square error (<10%) compared with force plate data [14,21]. One statistical limitation of these studies was that such correlation analyses do not reflect agreement between measurement systems, which was recommended [22]. Another limitation is that the 12-sensor layout’s validation [14] was based on vGRF without consideration of CoP. Other insoles with fewer sensors were proposed without validation [23] or based on limited data (e.g., peak values and visual curves) [13]. Extremely simplified insole layouts (<5 sensors) were used for specific purposes only (e.g., identification of arch types) but not for CoP analyses [24,25]. Therefore, there is no strong evidence supporting the validity of insoles comprising around 10 or fewer sensors. At a certain point, it may also be questionable as to whether further sensor number reduction significantly contributes to a practically relevant reduction in product costs and data amount.

The current literature demonstrated that one goal of scientific efforts is to reduce sensor numbers and facilitate affordable access to accurate data in clinical and therapeutic fields [15]. However, the minimal sensor number to accurately estimate CoP is unclear. The relationship between reduction in sensor number and measurement accuracy is required to identify an optimal compromise between sensor number and accuracy of CoP. Presuming that the relationship between sensor number and accuracy is not perfectly linear, an optimal compromise can be identified based on curve patterns in sensor number-accuracy plots (e.g., at a potential plateau or a local maximum).

The objectives of this study were to (1) assess the agreement and error in CoP data from multiple insole layouts comprising different sensor numbers compared with the complete and valid original layout and (2) determine a promising compromise between sensor number and CoP accuracy during different types of gaits. It was hypothesized that simplified sensor layouts can obtain valid CoP with small errors compared with the original layout. A secondary hypothesis was that fractal CoP direction and type of gait would affect accuracy.

## 2. Materials and Methods

### 2.1. Participants

Twelve active sports students (age: 24.6 ± 3.7 years, body height: 173.1 ± 6.8 cm, body mass: 68.5 ± 7.8 kg, Pedar-X insole size: 27–28), free of lower limb injuries, participated in this study and signed written consent. A-priori power analysis was conducted via G*Power 3.1.9.7 (Heinrich Heine Universität, Düsseldorf, Germany), resulting in statistical power 1 − β = 0.80 for detecting effect sizes of r ≥ 0.66 at a significance level of α = 0.05. This calculation was performed for the primary correlation-based test of this study. Data were handled anonymously and the study followed the Declaration of Helsinki. The institutional ethics committee approved the study (approval number: 201803HM002; date: September 2019).

### 2.2. Instruments

A Pedar-X insole (Novel Electronics Inc., Saint Paul, MN, USA) comprising 99 capacitive sensors was used at 100 Hz. The system was described and validated previously [10,11,12]. Following the instruction manual, the insoles were calibrated in a loading range of 20 to 600 kPa and a Novel’s TruBlu calibration device was used. CoP was calculated as follows:(1)$${\mathrm{CoP}}_{\mathrm{ML}}=\frac{\sum_{i=1}^{n}x_{i}\times {vGRF}_{i}}{vGRF},$$
(2)$${\mathrm{CoP}}_{\mathrm{AP}}=\frac{\sum_{i=1}^{n}y_{i}\times {vGRF}_{i}}{vGRF},$$
where n represented the total number of sensors, i was the single sensor, and x and y were the mediolateral (ML) and anteroposterior (AP) coordinates, respectively. CoP accuracy was not expected to be affected by variations in foot size, as suggested by previous research [26].

The participants’ speed in the movement direction was derived from a reflective marker (diameter: 15 mm) at the cervical vertebrae, captured by ten Vicon vero and two Vicon T40-S cameras (Vicon Ltd., Oxford, UK) at 200 Hz. Data were managed via Vicon Nexus 2.10.1 software.

### 2.3. Protocol

The laboratory provided the same type of shoes and socks to all participants. After familiarization, the participants completed three trials of straight walking, jogging, and running at self-selected constant speeds for several meters to allow for natural gait patterns. A single step per participant from the middle of each trial was used for further analyses to avoid a hierarchical data structure (i.e., multiple steps per participant) and the effects of acceleration and deceleration. The sequence of locomotion types was randomized and participants were free to take 1-min breaks between trials. Figure 1 depicts the procedure.

### 2.4. Pedar-X Layout

The original 99-sensor Pedar-X layout served as a valid reference layout. By extracting data from a reduced number of sensors during the identical trials, eight simplified layouts were simulated without repetition bias. These layouts (see Figure 2) were selected based on the previously mentioned literature on foot anatomy, gait kinetics, and previously used layouts [13,14,15,16,17,18,19,20,21,23]. To provide a few examples, the main pressure distribution across the heel and the entire width of the forefoot [17] recommended a 3-sensor placement at the heel and along the mediolateral axis of the forefoot. The importance of the hallux for balance [21] and most pressure (60%) being located at the heel across a large distribution area [17], as well as the emphasis on covering the heel with multiple sensors [14,15,16,21] suggested two additional sensors at the hallux and the heel for the 5-sensor layout. Previous 7-sensor layouts included 3 sensors at the forefoot and at least one sensor at the midfoot to cover the pathway of the CoP [13,23]. Previous insoles incorporating 12 and 13 sensors used additional sensors to cover the mediolateral axis of the toes, the mediolateral and anteroposterior axes of the midfoot, and a larger distribution area at the heel [14,16], which was gradually implemented in the current layouts comprising 9 and more sensors. From simple to complex layouts, the first priority was anatomical locations with high pressure, the second was alignment with the CoP trajectory, the third was the effects of arch types, and the fourth was increasing the resolution in high-pressure areas. Moreover, the exact layouts were used in previous research assessing the accuracy of vGRF [20]; their limitations in quantifying CoP need to be investigated in the current study.

### 2.5. Statistics

PASW Statistics 18.0 (SPSS Inc, Chicago, IL, USA) and Excel 365 (Microsoft Corporation, Redmond, WA, USA) were used for statistical analyses and visualization. Analyses were performed separately for the fractal components of CoP (i.e., ML and AP). A normal distribution of the respective data for statistical tests described below was checked via the Shapiro–Wilk test, skewness, and kurtosis. Data were presented as mean ± standard deviation (SD).

The instantaneous CoP data over the entire time series during ground contact was used for calculations of concordance correlation coefficients (CCC) [27] to assess the agreement between the original and each simplified sensor layouts in all conditions separately. This was implemented for each participant to avoid data clustering. The global mean CCC across participants and 95% confidence intervals (95% CI) were calculated for each locomotion type. CCC was chosen because it is a recommended inferential method to test agreement between two items [22]. Higher CCC values represent stronger agreement, which was desirable. Agreement was interpreted as poor (CCC < 0.90), moderate (0.90 ≤ CCC < 0.95), and substantial (CCC ≥ 0.95) [28].

For each participant and condition, the root mean square error (RMSE) of CoP over the time series between the reference and each simplified layout was calculated separately for ML and AP. RMSE was assessed in the context of the maximal range of CoP movement, derived from the original Pedar-X as the discrepancy between minimal and maximal fractal CoP values. RMSE was chosen because it is a frequently used expression of absolute error in validation studies [12,15]. Since RMSE is non-directional, the 95% CI of directional difference between the reference and simplified layouts was calculated, separately for fractal CoP, to test systematic error tendencies.

Friedman’s test was performed for differences in CCC and RMSE between sensor layouts, reported as chi-square ($\chi^{2}$) with degrees of freedom. Analysis of variance (ANOVA) with repeated measures was conducted to test differences between fractal CoP for CCC and differences between conditions for CCC, RMSE, and participants’ movement speeds. For post-hoc comparison, Bonferroni correction was manually applied, accounting for the actual number of conducted tests. Effect sizes of ANOVA results were presented as the partial eta square ($\eta_{p}^{2}$) and interpreted as trivial, small, medium, and large at the thresholds of 0.10, 0.25, and 0.40 [29]. Power was expressed as 1 − β. The significance level for all tests was set at *p* < 0.05. A promising compromise was derived based on the relative change in fractal CCC between layouts, normalized to the relative increase in sensor number, separately for each condition:(3)$${\Delta CCC}_{rel}=\frac{{CCC}_{i+2}-{CCC}_{i}}{\left(\frac{n_{i+2}-n_{i}}{n_{i}}\right)}$$
where *i* represents a layout, *i* + 2 is the layout with two additional sensors, and *n* is the number of sensors. The compromise was determined when improvement in ΔCCC_rel_ was reduced by at least twice the SD of CCC between previous layouts. The threshold of twice the SD was chosen as it is a common threshold in the field to identify outliers. Thus, this threshold allows for the identification of a reduction beyond the usual variation between previous layouts. The assessment was supported by the visual interpretation of RMSE development.

## 3. Results

The self-selected movement speed differed between walking (1.26 ± 0.16 m·s^−1^), jogging (2.34 ± 0.28 m·s^−1^), and running (3.18 ± 0.64 m·s^−1^) (*F*_(1.16,12.74)_ = 86.38, *p* < 0.001, $\eta_{p}^{2}$ = 0.89, 1 − β = 1).

During all three conditions, the agreement between the original Pedar-X and simplified sensor layouts varied across sensor layouts for CCC_ML_ ($\chi_{\left(7\right)}^{2}$ ≥ 46.16, *p* < 0.001) and CCC_AP_ ($\chi_{\left(7\right)}^{2}$ ≥ 34.94, *p* < 0.001). Differences across conditions were found in CCC_ML_ (*F*_(1.23,78.79)_ = 16.50, *p* < 0.01, $\eta_{p}^{2}$ = 0.70, 1 − β = 0.97) and CCC_AP_ (*F*_(1.10,7.72)_ = 44.34, *p* < 0.001, $\eta_{p}^{2}$ = 0.86, 1 − β = 1) and post-hoc comparison was displayed in Figure 3. All CCC means ± SD and 95% CI were presented in Table 1. A promising compromise via ΔCCC_rel_ was identified in layouts with 11 (walking_ML_ and jogging_AP_) and 13 (walking_AP_, jogging_ML_, running_ML_, and running_AP_) sensors.

During all three conditions, the error between the 99-sensor layout and simplified layouts (i.e., RMSE) varied across layouts for RMSE_ML_ ($\chi_{\left(7\right)}^{2}$ ≥ 57.69, *p* < 0.01) and RMSE_AP_ ($\chi_{\left(7\right)}^{2}$ ≥ 63.25, *p* < 0.001). Differences across conditions were found in RMSE_ML_ (*F*_(2,14)_ = 10.40, *p* < 0.01, $\eta_{p}^{2}$ = 0.60, 1 − β = 0.96) and in RMSE_AP_ (*F*_(2,14)_ = 14.27, *p* < 0.001, $\eta_{p}^{2}$ = 0.67, 1 − β = 0.99); post-hoc comparison was displayed in Figure 3. Figure 4 depicted all RMSE means ± SD and 95% CI, including a post-hoc comparison between layouts. The maximal range of CoP_ML_ and CoP_AP_ movement was 32 ± 3 mm and 191 ± 9 mm, respectively. The 95% CI of the directional difference in CoP [mm] between the reference and simplified layouts ranged from −3.7 to −0.9 (walking), −5.5 to −1.2 (jogging), and −6.5 to −2.2 (running) for ML and from 1.2 to 6.0 (walking), −12.1 to −1.5 (jogging), and −13.6 to −2.5 (running) for AP, where negative values represented the error in simplified layouts toward the medial and anterior directions.

## 4. Discussion

The main objective of this study was to investigate the agreement and error of CoP data between the original 99-sensor Pedar-X insole and various simplified sensor layouts during three different movement conditions. Walking, jogging, and running differed in self-selected movement speeds without any overlap of speed ranges. Speed variations within locomotion types were minor compared to the differences between types. Therefore, locomotion types were clearly distinguishable and the data suggested no influence of speed on the further analyses beyond the level of locomotion types, which have been analyzed. Differences between conditions were also observed in agreement (expressed as CCC) and error (expressed as RMSE). In both CoP_ML_ and CoP_AP_ and both CCC and RMSE, four out of four differences were documented between running and jogging, three out of four between jogging and walking, and one out of four between walking and running (Figure 3). This is in line with previous studies that reported different performances of identical pressure insoles during different tasks [16]. Most recently, locomotion-dependent accuracy of pressure insoles was noted for vGRF [20]. Overall, this supports the effect of locomotion type on agreement and error. Therefore, optimization strategies for sensor reduction as well as validation attempts should be movement-specific. This entails coaches and applied sports scientists paying attention to the validation specificity of products and being aware of the tasks in which they wish to use pressure insoles.

The largest RMSE in both fractal CoP were found during walking and the smallest RMSE during jogging; RMSE during running was larger than during jogging and was comparable with walking (Figure 3). Like RMSE results, CCC values of CoP_ML_ reflect the poorest performance during walking, comparable with running, and improved during jogging. The only exception where results during walking were not the poorest but were comparable with jogging and favorable over running was found in CCC values of CoP_AP_. In general, higher agreement and smaller errors were achieved in jogging than in walking and running.

Across all sensor layouts, CCC values of CoP_AP_ tended to be higher than the ones of CoP_ML_ in all conditions. A similar trend was observed previously where correlation was stronger in CoP_AP_ than in CoP_ML_ [19]. Absolute RMSE was larger for CoP_AP_ than for CoP_ML_ in all conditions. The previously reported RMSE for CoP_ML_ (4–11 mm) [14,21] was comparable with the current results. For CoP_AP_, previous RMSE (13–17 mm) [14,21] were smaller than current results in walking, comparable with most of the current layouts in running, and larger than most layouts in jogging. However, to assess the relevance of these absolute errors, the maximal range of CoP movement covered by the insole in both directions should be considered. In doing so, the relative errors across sensor layouts and conditions were smaller in RMSE_AP_ (3.66–17.80%) than in RMSE_ML_ (9.36–40.63%). This suggests that simplified sensor layouts may be more accurate for the assessment of CoP changes in AP than in ML. Practically, this observation may be of lesser concern when the CoP pathway primarily follows the AP direction (e.g., linear gaits). However, attention is advised in tasks where ML sway dominates the CoP pathway (e.g., lateral changes of direction or static balance tasks with lateral disturbances).

The differences in both CCC and RMSE across sensor layouts corroborate that the accuracy of sensor layouts depends on the number of sensors. Some layouts perform better than others, which justifies the purpose of the study to identify a good compromise between accuracy and sensor number. Therefore, it is reasonable to consult ΔCCC_rel_ results in combination with the assessment of individual performances of sensor layouts (Table 1 and Figure 4).

In running, almost all CCC results were poor. Only the 17-sensor layout showed at least moderate agreement with the reference layout for CoP_ML_ (0.91 ± 0.12). RMSE also tended to be large in running. Whereas complex layouts yielded acceptable RMSE in the ML direction (e.g., 3 mm), even the best result in the AP direction (i.e., 14 ± 15 mm) represented an error as large as 7% of the total CoP_AP_ distance covered. Although the absolute RMSE was comparable with previous findings [14,21], it is questionable if this is acceptable for accurate measurements. None of the simplified layouts should be used for scientific investigations of CoP during running. In practical fields or other scenarios where decreased accuracy is sufficient, 13 sensors can be recommended because CCC approximately reached the moderate benchmark and RMSE_ML_ was improved compared with the 11-sensor layout. It is unclear as to whether results during running may have been negatively affected by the sampling frequency. If so, higher sampling frequencies may improve accuracy during running. However, following this argument, the highest accuracy should have been reported in the task with the slowest CoP movement (i.e., in walking). This was not the case as accuracy was higher in jogging than in walking. Therefore, although some may assume an influence of sampling frequency on accuracy specifically in running, the findings demonstrated that sampling frequency was not the only explanation of effects between tasks.

In walking, multiple sensor layouts obtained moderate to substantial agreement. However, all these results except for one were found in the AP direction. All CCC results in the ML direction were poor except for the 17-sensor layout. Therefore, the magnitude of potential error in the ML direction should be considered when using simplified layouts in walking. An optimal compromise range was quantified between 11 and 13 sensors. However, this was not corroborated by the constantly decreasing RMSE as the number of sensors increased. The deficient agreement in ML direction and large RMSE across all sensor layouts do not recommend simplified sensor layouts in settings where high accuracy is required (e.g., scientific measurements).

In jogging, the smallest errors and best agreement results across sensor layouts were reported. Layouts comprising 11 or more sensors achieved substantial agreement in at least one of the two fractal CoP data sets. The 13-sensor layout seemed to stand out because both CoP_ML_ and CoP_AP_ substantially agreed with the reference layout and even the lower bound 95% CI of CoP_AP_ was above the benchmark of substantial agreement. The first aspect is important for accurate measurement of resultant CoP, which is usually more relevant in CoP assessment than a single fractal component. The second aspect expresses the great confidence that the true agreement is, indeed, substantial, which was recommended previously [28]. A promising compromise range was quantified via ΔCCC_rel_ between 11 and 13 sensors. RMSE tended to improve until the 13-sensor layout where it stabilized eventually. Considering CCC and RMSE trends conservatively, the 13-sensor layout seemed to be a promising compromise between accuracy and sensor number.

Considering the major improvements in CoP accuracy from the 5-sensor to the 11-sensor layout, it is worth pointing out that the biggest difference among these layouts was in the midfoot. In particular, the 7-sensor layout introduced the first midfoot sensor. The 9-sensor layout added a second midfoot sensor along the AP direction and, therefore, improved the resolution along the CoP pathway during gaits. The 11-sensor layout added two more midfoot sensors to increase the ML coverage. The observed improvement in this sensor number range supports the relevance of the midfoot region for accurate CoP estimation. This may be considered in sports and therapy use of insoles for CoP monitoring because commercial simplified insoles show differences with respect to midfoot coverage.

For practical and therapeutical contexts, requirements of measurement accuracy may be different than for scientific investigations. Therefore, it should be noted that interpretation benchmarks are not set in stone (see different interpretations [28,30]). For example, a more liberal interpretation approach closer to other correlation coefficients accepts values >0.80 as strong [30] and may be considered alternatively for applications in practical scenarios. It may be argued that a strong correlation (i.e., >0.80) [30] between an affordable device and a valid reference may be sufficient if it enables practical applications. In such scenarios, the 11-layout may be used as the simplification with the least sensors that provided over 80% agreement in both fractal CoP across all investigated conditions. Nevertheless, the 13-sensor layout can be seen as the better compromise because of the mentioned significant improvements compared with the 11-sensor layout.

Finally, errors were systematic, as shown by the directional difference between reference and simplified layouts. All the 95% CI of these differences suggested clear error tendencies in simplified layouts toward a particular direction, mostly anterior and medial. It was beyond the scope of this study to examine the error source. However, it is reasonable to presume that systematic errors may be solved more easily in the future than non-systematic errors and, therefore, the accuracy of simplified layouts may be improved further.

Limitations of the current study include the limited types of movements (i.e., walking, jogging, and running) and the lack of comparison of different sensor sizes as well as layouts with the same number of sensors but different sensor placements. The effect of sensor size on CoP accuracy is unclear. Insoles with reduced sensor numbers allow for incorporating larger sensors. Therefore, it may be a justified consideration in future research to experiment with various sensor sizes. Sensor placement influences the accuracy of pressure insoles [21]. This effect of placement may be larger for CoP than for vGRF because CoP is determined by the location of the applied force. This is supported by the previous observation that two layouts with different sensor placement correlated comparably with the reference layout for vGRF (coefficients: 0.90 ± 0.08 and 0.91 ± 0.08) but the discrepancy increased for CoP_ML_ (0.91 ± 0.04 and 0.95 ± 0.02) and CoP_AP_ (0.97 ± 0.02 and 0.99 ± 0.01) [21]. Although the current study thoroughly defined placement based on evidence-based concepts and previously tested layouts, improvements through specific optimization strategies [10] may be plausible. A previous study [10] identified ideal sensor locations for vGRF estimation during walking via regression analysis. Such or similar approaches could also be used for CoP. However, it requires cross-validation and, therefore, a large sample size to achieve high levels of agreement between estimated and measured data. The current study provided a promising range of sensor numbers for accurate CoP measurements across different gaits; its goal was not to determine the optimal sensor placement. For such a purpose, future research is advised to compare various sensor placements within the range of sensor numbers recommended in the current study.

## 5. Conclusions

This study investigated the effect of sensor numbers in simplified pressure insole layouts on the measurement accuracy of CoP. Simplified sensor layouts tended to not be recommended for walking and running in scientific investigations and when high accuracy is warranted. In the absence of more accurate products, simplified layouts with at least 11 sensors achieved over 80% agreement with the reference layout and, therefore, may be suitable for practical and therapeutic applications. In jogging, layouts comprising 11 or more sensors achieved a substantial agreement with the reference layout and small errors, both in a practical context and compared with previous studies. Overall, the 13-sensor layout provided a promising compromise between accuracy and sensor number across locomotion types. Future research on the optimization of simplified sensor layouts may focus on layouts integrating 11 to 13 sensors.

## Figures and Tables

**Figure 1 sensors-24-04918-f001:**
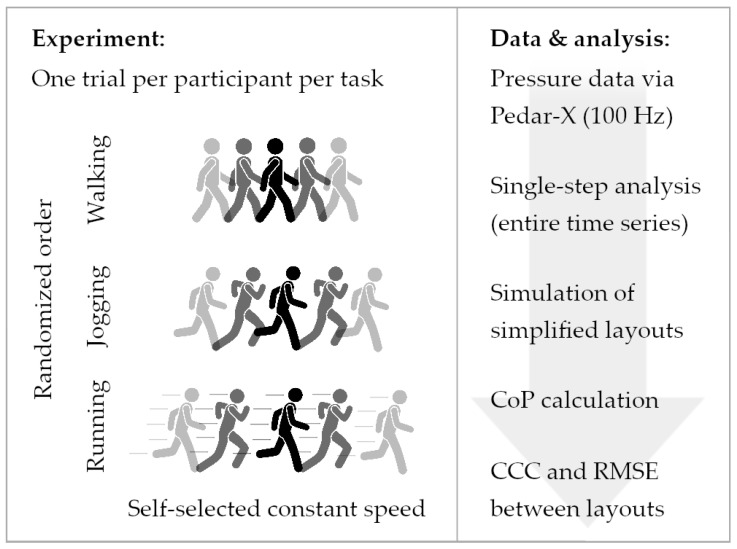
Experimental and data handling procedure. CoP: center of pressure, CCC: concordance correlation coefficients, RMSE: root mean square error.

**Figure 2 sensors-24-04918-f002:**
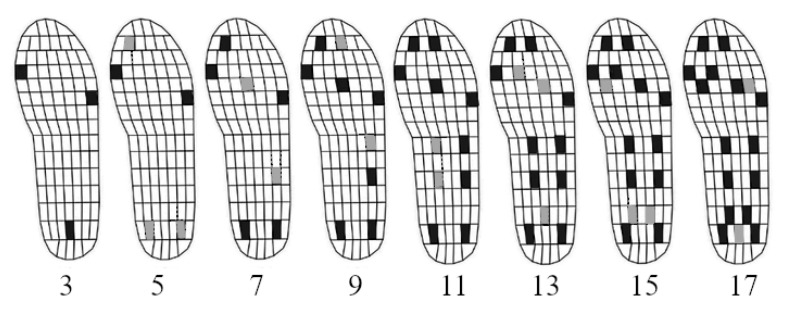
Simplified layouts comprising different numbers of sensors [18]. Black fields: involved sensors; grey fields: sensors that were added compared to the more simplified layout; 3: distal first metatarsal, lateral fifth metatarsal, and central heel; 5: 3 plus hallux, and central heel replaced by the medial and lateral heel; 7: 5 plus central metatarsal and proximal lateral midfoot; 9: 7 plus central tarsal and distal lateral midfoot; 11: 9 plus proximal and distal midfoot; 13: 11 plus distal heel and central metatarsal replaced by two adjacent metatarsals; 15: 13 plus proximal first metatarsal and distal heel replaced by two adjacent distal heels; 17: 15 plus fifth medial metatarsal and central heel.

**Figure 3 sensors-24-04918-f003:**
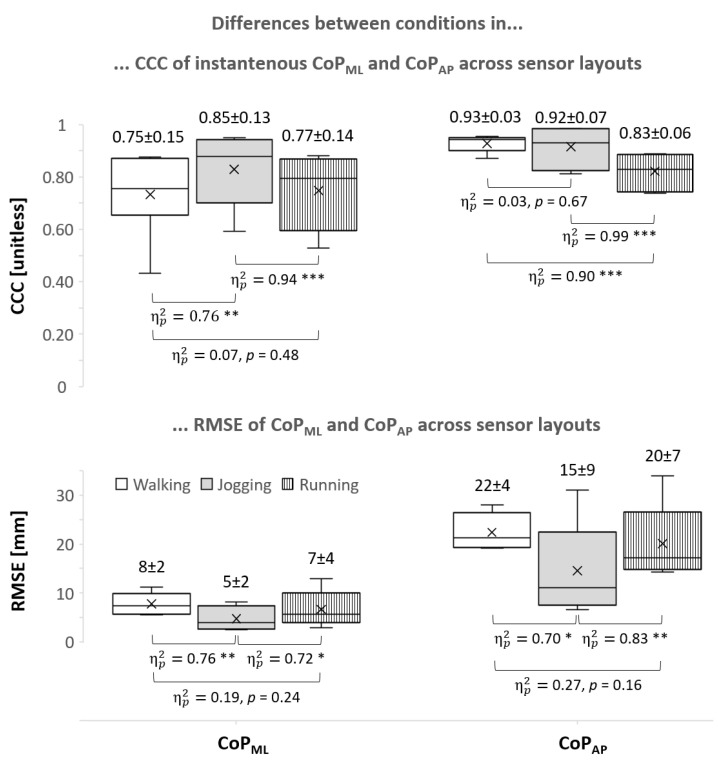
Boxplots of concordance correlation coefficients (CCC) and root mean square errors (RMSE) of fractal center of pressure (CoP_ML_ and CoP_AP_) including mean ± standard deviation and post hoc comparison between conditions. *: *p* < 0.05; **: *p* < 0.01; and ***: *p* < 0.001 after Bonferroni correction.

**Figure 4 sensors-24-04918-f004:**
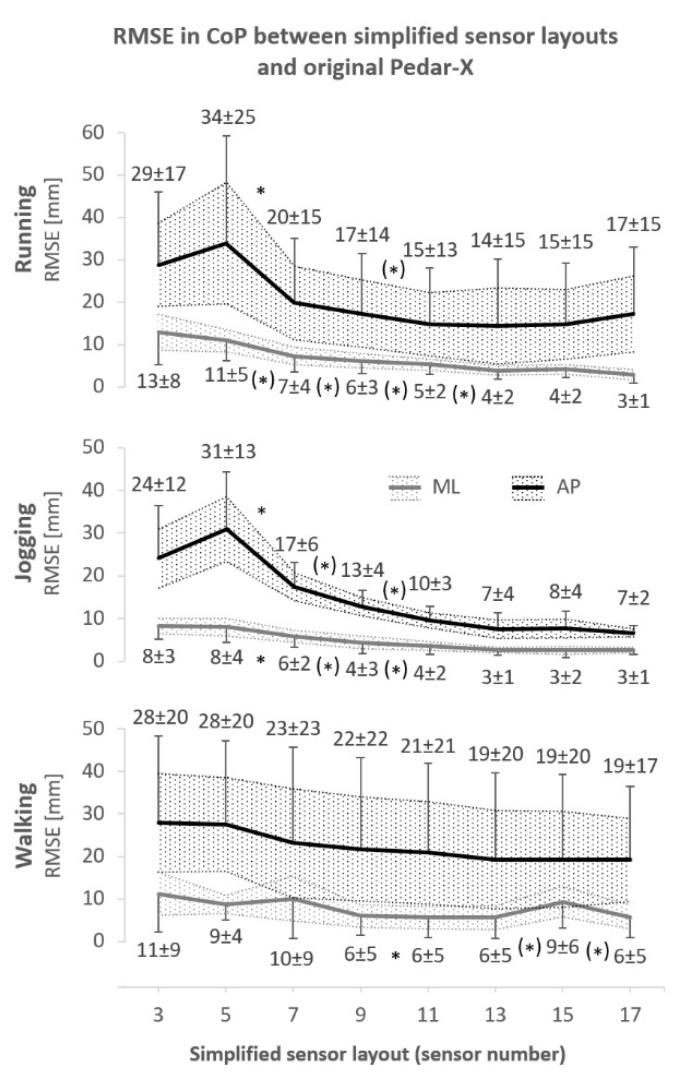
Root mean square errors (RMSE) of mediolateral (ML) and anteroposterior (AP) center of pressure (CoP) between the original and simplified sensor layouts as mean ± standard deviation and 95% confidence intervals as dotted areas. *: significant (*p* < 0.05) post hoc comparison between the adjacent sensor layouts after Bonferroni correction; (*): significant only without Bonferroni correction.

**Table 1 sensors-24-04918-t001:** Agreement of instantaneous center of pressure (CoP) in the mediolateral (ML) and anteroposterior (AP) directions between simplified sensor layouts and original Pedar-X for all layouts and all conditions, as concordance correlation coefficients’ (CCC) mean ± standard deviation and 95% confidence intervals (95% CI).

		Sensor Layout (Sensor Number)							
		3	5	7	9	11	13	15	17
**Walking**	**CoP_ML_**	0.43 ± 0.36 (0.23–0.64)	0.65 ± 0.20 (0.54–0.77)	0.73 ± 0.38 (0.52–0.94)	0.81 ± 0.10 (0.75–0.86)	0.88 ± 0.08 (0.83–0.92)	0.87 ± 0.11 (0.81–0.93)	0.76 ± 0.45 (0.50–1)	0.90 ± 0.10 (0.84–0.95)
	**CoP_AP_**	0.87 ± 0.14 (0.79–0.95)	0.90 ± 0.12 (0.83–0.97)	0.93 ± 0.14 (0.85–1)	0.94 ± 0.13 (0.87–1)	0.94 ± 0.13 (0.87–1)	0.95 ± 0.11 (0.90–1)	0.95 ± 0.11 (0.89–1)	0.93 ± 0.11 (0.87–1)
**Jogging**	**CoP_ML_**	0.59 ± 0.16 (0.50–0.69)	0.70 ± 0.13 (0.63–0.77)	0.84 ± 0.08 (0.80–0.88)	0.88 ± 0.08 (0.84–0.92)	0.90 ± 0.07 (0.86–0.94)	0.95 ± 0.05 (0.92–0.98)	0.94 ± 0.05 (0.91–0.97)	0.96 ± 0.03 (0.94–0.97)
	**CoP_AP_**	0.83 ± 0.21 (0.71–0.94)	0.81 ± 0.16 (0.72–0.90)	0.91 ± 0.11 (0.85–0.98)	0.93 ± 0.13 (0.86–1)	0.96 ± 0.06 (0.93–1)	**0.98 ± 0.02** **(0.97–0.99)**	**0.98 ± 0.02** **(0.97–1)**	0.96 ± 0.06 (0.93–0.99)
**Running**	**CoP_ML_**	0.53 ± 0.22 (0.40–0.66)	0.60 ± 0.21 (0.48–0.72)	0.74 ± 0.18 (0.64–0.84)	0.79 ± 0.16 (0.70–0.88)	0.83 ± 0.12 (0.76–0.90)	0.88 ± 0.12 (0.82–0.95)	0.87 ± 0.13 (0.80–0.94)	0.91 ± 0.12 (0.85–0.98)
	**CoP_AP_**	0.74 ± 0.26 (0.59–0.89)	0.74 ± 0.30 (0.57–0.91)	0.81 ± 0.29 (0.64–0.97)	0.83 ± 0.28 (0.67–0.98)	0.86 ± 0.23 (0.74–0.99)	0.89 ± 0.22 (0.76–1)	0.89 ± 0.21 (0.77–1)	0.87 ± 0.21 (0.75–0.99)

Note: Fields were marked as small-sized if mean CCC < 0.80, regular-sized if mean CCC ≥ 0.80, grey background if mean CCC ≥ 0.95, and bold if lower bound 95% CI CCC ≥ 0.95.

## Data Availability

The original contributions presented in the study are included in the article/Appendix A, further inquiries can be directed to the corresponding author.

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
