# Peer review of "Center of Pressure Measurement Accuracy via Insoles with a Reduced Pressure Sensor Number during Gaits"

_sensors, 2024, doi:10.3390/s24154918_

Round 1

Reviewer 1 Report

Comments and Suggestions for Authors

This manuscript describes the assessment of various reduced pressure sensor layouts in their accuracy of approximating correct center of pressure readings. It is overall well written and makes an interesting contribution to the field. Yet, it may be further improved by clarifying a few aspects as detailed below.

The general idea seems straightforward – of course, higher resolution (= larger number of sensors) results in better COP estimation. The motivation for the study could be strengthened if the rationale for hypothesizing otherwise was clarified.

Line 63 – this statement is confusing and may need more context/elaboration

L87 – A worthwhile point of discussion, to be included here or elsewhere, would be the size of the sensors. Depending on the function principle of the sensor, it is possible to have fewer but larger sensors with likely implications for the accuracy of the COP and other measurements.

L97 – the number of participants and other sample characteristics should be reported under results. Instead, consider including information on the a-priori power analysis here.

L125 – is there a possibility of sensor cross-talk that would affect the validity of this method (e.g., related to the wiring)?

L164 – the definition for the moderate category should be clarified to avoid overlap with the poor category (below 90)

L166 – I believe this alternative interpretation would be better brought up in the Discussion chapter. It diminished the rigor of the methods here.

L177 – Can it be clarified what hypothesis was investigated with those significance tests? I was under the impression that a correlation analysis would be most appropriate to arrive at some sort of plot that illustrates the optimal tradeoff between sensor number and accuracy.

L189 – This appears to be a somewhat random criterion. Is there any prior work that could support this choice?

L192 – Much like for velocity numbers, it would be recommendable to include leading zeros when reporting numbers <1 (e.g., p values)

Much of the text within the Results chapter is quite difficult to read, due to the multitude of parameters reported for each analysis. This reviewer would prefer a presentation in graph or table form whenever possible (Figure 3 is very effective). In any event, information should not be repeated between text and graphs/tables.

The Discussion chapter reads very well and covers all the relevant content. In acknowledging the limitations of not modulating sensor positions, it may be discussed what it would take to run a sort of factorial analysis that compares a large number (all) of the possible sensor configurations against the 99-sensor gold standard. This approach would seem to most definitively answer the research question.

Author Response

Comment: This manuscript describes the assessment of various reduced pressure sensor layouts in their accuracy of approximating correct center of pressure readings. It is overall well written and makes an interesting contribution to the field. Yet, it may be further improved by clarifying a few aspects as detailed below.

Response: We thank the reviewer for the time committed to rigorous review. We appreciate the positive impression and the comments, which helped us improve the manuscript.

Comment: The general idea seems straightforward – of course, higher resolution (= larger number of sensors) results in better COP estimation. The motivation for the study could be strengthened if the rationale for hypothesizing otherwise was clarified.

Response: We are glad that the idea was perceived as straightforward. To further strengthen the motivation of the study, we added a clarification of the rationale for the hypothesis to the second last paragraph of the introduction section.

Comment: Line 63 – this statement is confusing and may need more context/elaboration

Response: As per reviewer’s suggestion, we added more context about the referenced study and how those findings relate to the subsequent statement.

Comment: L87 – A worthwhile point of discussion, to be included here or elsewhere, would be the size of the sensors. Depending on the function principle of the sensor, it is possible to have fewer but larger sensors with likely implications for the accuracy of the COP and other measurements.

Response: We agree that sensor size affects the measured data, maybe even the accuracy of the derived CoP. However, we did not find any evidence on whether sensor size would systematically affect CoP accuracy and whether a larger size would increase or decrease accuracy (if there was an effect). Therefore, we feel we cannot add reliable information about this aspect, and adding assumptions about sensor size, which was beyond the scope of the study, could probably not enrich the discussion or conclusions. However, we added a mention of sensor size to the limitations (last paragraph of the discussion section).

Comment: L97 – the number of participants and other sample characteristics should be reported under results. Instead, consider including information on the a-priori power analysis here.

Response: As per reviewer’s suggestion, we added information on the a-priori power analysis as well as sample characteristics. We prefer keeping the number of participants and the newly added sample characteristics in the chapter “2.1. Participants” as this is the norm in the authors’ research field, but we are fine with moving this information to the results section if the reviewer and editor find that more appropriate for the journal’s readership.

Comment: L125 – is there a possibility of sensor cross-talk that would affect the validity of this method (e.g., related to the wiring)?

Response: Cross-talk exists, and there are concrete numbers documented for the Pedar-X system specifically [Saggin, B., Scaccabarozzi, D., & Tarabini, M. (2013). Metrological performances of a plantar pressure measurement system. IEEE Transactions on Instrumentation and Measurement, 62(4), 766-776]. The effect on the current method seems marginal to us because there are two aspects to the regular cross-talk that counteract and neutralize each other when using our method. On the one hand, cross-talk increases measured pressure at adjacent sensors (11 N at an unloaded sensor per 100 N of load at an adjacent sensor to the left and right, and 5 N per 100 N to the top and bottom). So, this aspect of the cross-talk increases pressure on a sensor in our simulated layout due to mechanically existing sensors that were not included in the simulation. However, the second aspect is that loaded sensors systematically underestimate loads (81.3 N measured per 100 N of load). This is probably due to calibration that takes the first aspect of cross-talk into account. Regardless the reason, that means for our method that the first and the second aspect may neutralize each other. We hope that this explanation solves the reviewer’s concern. We did not add a complete elaboration to the full-text as we think that would require a quite wordy explanation and yet may be more confusing than helpful for a large part of the readership.

Comment: L164 – the definition for the moderate category should be clarified to avoid overlap with the poor category (below 90)

Response: As per reviewer’s suggestion, the moderate category was changed to .90≤CCC<.95, and “CCC” was added to the poor and substantial categories to match the new format.

Comment: L166 – I believe this alternative interpretation would be better brought up in the Discussion chapter. It diminished the rigor of the methods here.

Response: Thank you. We found a suitable location in the third last paragraph of the Discussion section for the alternative interpretation.

L177 – Can it be clarified what hypothesis was investigated with those significance tests? I was under the impression that a correlation analysis would be most appropriate to arrive at some sort of plot that illustrates the optimal tradeoff between sensor number and accuracy.

Response: The reviewer’s impression of correlation analysis, illustrating an optimal tradeoff is correct. This sort of analysis was conducted as described by the reviewer (CCC is correlation-based; RMSE was illustrated as a plot, but for CCC we used the table to be able to provide precise 95% CI which matters for the benchmark interpretation). The mentioned significance tests were only additions to investigate moderating effects. For clarification, we propose adding a secondary hypothesis (see at the end of the introduction section) that is tested through the mentioned tests.

Comment: L189 – This appears to be a somewhat random criterion. Is there any prior work that could support this choice?

Response: We assume the reviewer refers to using twice the SD to determine the compromise (“when improvement of ΔCCCrel was reduced by at least twice the SD of CCC between previous layouts”). We looked for existing approaches that would fit our purpose and case but could not find one because other studies did not involve such a sort of quantification. However, we think it is better to provide a quantified approach in addition to the qualitative assessment than not providing it. We chose twice the SD because it is a very common threshold for identifying outliers. A reduction of more than twice the SD means that such change is beyond regular variation up to this point. We added an explanation to the manuscript.

Comment: L192 – Much like for velocity numbers, it would be recommendable to include leading zeros when reporting numbers <1 (e.g., p values)

Response: As per reviewer’s suggestion, we added leading zeros to all reported numbers <1 throughout the manuscript including figures and tables.

Comment: Much of the text within the Results chapter is quite difficult to read, due to the multitude of parameters reported for each analysis. This reviewer would prefer a presentation in graph or table form whenever possible (Figure 3 is very effective). In any event, information should not be repeated between text and graphs/tables.

Response: We revised the results section and found a paragraph that was repeated by mistake and some redundancies between text and figure 2. We removed those redundancies from the text. In addition, we implemented few changes in phrasing and sentence structure to facilitate readability of the remaining text.

Comment: The Discussion chapter reads very well and covers all the relevant content. In acknowledging the limitations of not modulating sensor positions, it may be discussed what it would take to run a sort of factorial analysis that compares a large number (all) of the possible sensor configurations against the 99-sensor gold standard. This approach would seem to most definitively answer the research question.

Response: We added a brief elaboration of that idea of sensor positions to the last paragraph in the discussion section. We referred to a previous study and provided our point of view on what it would take (i.e., 1. Cross-validation, 2. Large sample size, 3. Agreement testing).

Reviewer 2 Report

Comments and Suggestions for Authors

This is an interesting article investigating how the number of sensors affects the results of pressure-sensing insoles.

The reviewer has the following comments: 

·         In the Abstract, it is crucial to define the CCC and RMSE, as this will provide a clear understanding of the research and keep the audience engaged.

·         It is essential to include a figure that illustrates the actual experiment setup, providing a clear visual aid for understanding the research.

·         When making an insole with fewer sensors, the size of each sensor would be larger, like, e.g., OPENGO, the sensor selected in Fig.1 is a small sensor. Should the researcher consider including a few nearby sensors to form the new sensor location? Or compare both cases as another finding of the study?

·         Maybe obvious, but would it be better to say what is considered better? Large or small CCC?

·         Why, in Fig. 3, 5 sensors give worse RMSE compared to 3 sensors?

Comments on the Quality of English Language

n/a

Author Response

Comment: This is an interesting article investigating how the number of sensors affects the results of pressure-sensing insoles.

Response: We appreciate the reviewer’s positive impression as well as the comments, which helped us strengthen the manuscript.

Comment: In the Abstract, it is crucial to define the CCC and RMSE, as this will provide a clear understanding of the research and keep the audience engaged.

Response: We added the full wording instead of only “concordance” and “errors” in front of the two abbreviations.

Comment: It is essential to include a figure that illustrates the actual experiment setup, providing a clear visual aid for understanding the research.

Response: The experimental setup is very straightforward. As per reviewer’s suggestion, we created a new figure (i.e., Figure 1) to provide visual aid.

Comment: When making an insole with fewer sensors, the size of each sensor would be larger, like, e.g., OPENGO, the sensor selected in Fig.1 is a small sensor. Should the researcher consider including a few nearby sensors to form the new sensor location? Or compare both cases as another finding of the study?

Response: Sensor size may be a relevant topic to study in the future. We are not aware of any evidence clearly indicating that sensor size affects CoP accuracy, let alone whether such a potential effect would support the use of larger or smaller sensors. Therefore, we lack reliable information to integrate meaningful considerations into our interpretations, and sensor size was beyond the scope of the current study. However, we agree that the reviewer made a justified point, and we addressed this aspect in the limitations (see: last paragraph in the discussion section).

Comment: Maybe obvious, but would it be better to say what is considered better? Large or small CCC?

Response: As per reviewer’s suggestion, we added a clarification to the second paragraph of 2.5 Statistics where CCC was explained.

Comment: Why, in Fig. 3, 5 sensors give worse RMSE compared to 3 sensors?

Response: With a new figure introduced, the reviewer’s comment refers to what is now Figure 4. First, this observation applies only to the AP fractal of CoP and only to jogging and running. Second, the difference was not significant. However, we share the surprise of the reviewer and agree that this trend is counterintuitive. We believe that those particular results have not been very reliable, which is supported by the large standard deviations observed in those results. Especially for jogging, where the trend seems stronger than in running, the standard deviations of 12 (3 sensors) and 13 (5 sensors) are twice as large as the second largest other SD (6 in the 7-sensor layout) and three times as large as the third and fourth largest other SD. Due to this imprecision, we cannot presume the true value and should not trust those particular mean values too much. CCC results do also not support that the 5-sensor layout would perform worse than the 3-sensor layout. Since those two layouts do not play a relevant role in our discussion and also do not for practical implications, we did not add such elaboration to the discussion section.